# Development and Validity of the Japanese Version of the Questionnaire on Factors That Influence Family Engagement in Acute Care Settings

**Makoto Tsukuda** [1],*, **Yoshiyasu Ito** [1], **Shota Kakazu** [2], **Katsuko Sakamoto** [3] **and Junko Honda** [1]

1 College of Nursing Art and Science, University of Hyogo, 13-71 Kitaoji-cho, Akashi 673-0021, Japan; yoshiyasu_ito@cnas.u-hyogo.ac.jp (Y.I.); junko_honda@cnas.u-hyogo.ac.jp (J.H.)
2 Graduate School of Nursing Sciences, St. Luke's International University, 10-1 Akashi-cho, Chuo-ku, Tokyo 104-0044, Japan; shota@slcn.ac.jp
3 Nursing Department, Hyogo Prefectural KOBE Children's Hospital, 1-6-7 Minatojima Minamimachi, Chuo-ku, Kobe 650-0047, Japan; kachi.o.o.dolphin@gmail.com
* Correspondence: makoto_tsukuda@cnas.u-hyogo.ac.jp; Tel./Fax: +81-78-925-9419

**Abstract:** There exists an international consensus on the importance of family-centered care (FCC) in intensive care settings and the evaluation of collaboration between nurses and families; however, FCC is currently practiced blindly in Japan. In this study, we developed a Japanese version of the questionnaire, Factors that Influence Family Engagement (QFIFE-J) and examined its reliability and validity. A web-based survey was conducted with 250 nurses working in the intensive care unit (ICU). Exploratory and validatory factor analyses were used to ascertain factor validity. Criterion-related validity was tested using correlation analysis with the ICU Nurses' Family Assistance Practice Scale. Internal consistency and reproducibility were verified for reliability. Following exploratory and confirmatory factor analyses, a 15-item measure emerged comprising four factors: "ICU environment", "nurses' attitudes", "nurses' workflow", and "patient acuity". Confirmatory factor analyses showed a generally good fit. Cronbach's α for the overall scale was 0.78, indicating acceptable internal consistency. The intraclass coefficient for test–retest reliability was 0.80. It was found that the QFIFE-J was reliable and valid and may help determine the factors that promote or inhibit FCC. Additionally, this study has also clarified the current status and family support related issues in ICUs in Japan.

**Keywords:** QFIFE; ICU nurses' family assistance practice scale; exploratory and validatory factor analyses; family-centered care; Japanese version

## 1. Introduction

Of late, intensive care technology has made dramatic progress, and various efforts have been made worldwide to focus on long-term goals such as daily life and social reintegration after discharge from the intensive care unit (ICU)—rather than short-term goals such as saving lives in the ICU. The background of this increased focus on long-term goals is the recognition of the fact that not only physical problems, but also mental disorders, including cognitive and emotional problems, may worsen long-term prognosis after discharge of critically ill patients from the ICU. Currently, this is termed Post-intensive Care Syndrome (PICS) [1].

The ICU provides advanced medical care, and patients admitted are often in life-threatening situations, causing sudden physical and mental stress, anxiety, confusion, and fatigue among their family members [2]. Uncertainty about such life crises, an unfamiliar hospitalization environment, and inadequate communication with healthcare providers can significantly affect families' interest in patient care and their coping abilities [3].

Mental disorders such as anxiety, depression, and complicated grief in the family that occur after the patient's admission to the ICU have gradually started gaining focus and are

now recognized as PICS-Family (PICS-F). It is an urgent issue to consider both the patient and the family after ICU admission as objects of care as in PICS.

Recently, academic focus on PICS and the PICS-F for intensively treated patients and their families has been driven by improved life-saving rates and ICU outcomes. It is largely because, in the ICU setting, the emphasis is on the patient and their family, rather than on the physician and disease, which leads to better outcomes. International guidelines supporting family-centered care (FCC) have been developed, and advanced practices are required to improve care and outcomes for patients and their families [4]. However, studies have established that many of these practices do not adequately identify nurses' perceptions of care for patients' families [4,5]. Moreover, research concerning both patient's families and nurses have reported various barriers to patient care, including the presence of patients' families in the ICU [6–8]. However, studies have demonstrated the benefits of the patient's family's involvement in patient care in the ICU; and the current trend requires the promotion of family care participation in critical areas in Japan [9,10]. Nurses must consider the facilitators and barriers that they face regarding family participation in patient care to promote collaborative partnerships among patients, families, and critical care nurses.

In Japan, an assessment tool for PICS was developed by Harumi et al. [11], which focused on assessing patients' PICS. A new scale is thus needed to assess factors related to collaboration between the nurses and patients' families. Hetland et al. [10] developed the Questionnaire on Factors that Influence Family Engagement (QFIFE)—a reliable questionnaire to investigate the facilitators and barriers faced by nurses working in the ICUs in the United States regarding family involvement in patient care and their associations [12]. This study developed and tested the reliability and validity of the Japanese version of the QFIFE (QFIFE-J).

## 2. Materials and Methods

### 2.1. Design

A cross-sectional design was employed, and a psychometric validation study was conducted using a web-based questionnaire.

### 2.2. Materials

2.2.1. Questionnaire on Factors That Influence Family Engagement

The QFIFE was developed in the United States as an acute care family nursing instrument to assess nurses' perceptions for involving families in the care of the critically ill [7]. The degree of barriers to FCC can then be assessed by the nurses' perceptions. This is because nurses' attitudes are important in promoting FCC. The QFIFE has 15 items comprising four subscales: (1) ICU environment (items 1–5), (2) Patient acuity (items 6–7), (3) Nurses' workflow (items 8–10), and (4) Nurses' attitudes (items 11–15). Each item can be rated on a six-point Likert scale. Higher item scores indicate a factor that facilitates patient-family involvement, while lower scoring items suggest a potential barrier factor for patient-family involvement. The internal consistency reliability of the QFIFE total scores and subscales is well-established with Cronbach's alpha of 0.73, 0.77, 0.74, and 0.83 for the ICU environment, Patient acuity, nurses' workflow, and Nurses' attitudes, respectively.

2.2.2. The Japanese Version of the Questionnaire on Factors That Influence Family Engagement

Intensive care nurses and quantitative research specialists were involved in the development of the Japanese version of the QFIFE. Discussions were also held with the original authors, as translations of individual items needed to be semantic rather than literal to ensure conceptual and linguistic equivalence. Thereafter, the Japanese version was retranslated into the English version. At this time, the first person in charge of the translation compared the back-translation with each other and with the English version. The wording of the item was then revised to consider the differences between the back-translation and the English version, and the revised Japanese was then retranslated to English by two

independent translators. After back-translation, the translated and back-translated scales were evaluated for semantic and conceptual aspects of the scales by bilingual translators and members of the re-study group. Finally, surface validity was confirmed by several nurses. The respondents were asked to ensure that all 15 questions were written in a clear, concise, and understandable manner. After checking the appropriateness of terminology and correcting spelling errors, a final version of the Japanese scale was created, which was named QFIFE-J.

### 2.3. Measures

2.3.1. Participants

This survey, which was conducted via an Internet-based questionnaire, was com-missioned by Rakuten Insight, Inc. (https://insight.rakuten.co.jp/en/, accessed on 9 June 2022). Approximately 6200 nurses working in ICUs are registered with Rakuten Insight, Inc. Survey requests were sent to ICU nurses who were registered with Rakuten Insight and who met the eligibility criteria of working full-time in the hospital ICUs. Associate nurses/nursing assistants and midwives were excluded. To validate a survey, a sample size that is at least seven times larger than the number of items it contains and an absolute number of at least 100 are required [13]. Of the nurses to whom the survey request was sent, those willing to participate were selected; the sampling continued until 250 nurses were recruited. To verify data collection, including reproducibility, two surveys were conducted to examine test–retest reliability. Two weeks after the first survey, a second survey was administered to all participants of the first survey. Data were collected between February to March 2022 in Japan.

2.3.2. Data Collection Procedures and Instruments

The questionnaire (the first survey) comprised (1) demographic characteristics, (2) the QFIFE-J, and (3) the Family Assistance Practice Scale for ICU Nurses [14].

1.  Participants' demographic characteristics. Data on participants' age, sex, educational background, years of clinical experience, years of critical care experience, certification, job position, number of beds in the hospital where the participant works, and type of unit were collected.
2.  The QFIFE-J. The QFIFE, developed by Hetland et al. [12], was used in the research survey by double-back-translation into Japanese, with the permission of the original creator. The six-question method, in which the answers to each question ranged from "strongly agree" to "completely disagree", was used to investigate the experiences and thoughts of the research collaborators.
3.  Family Assistance Practice Scale for ICU Nurses [15]. The Family Assistance Practice Scale for ICU Nurses was used to validate the criterion-related validity. This scale comprises four factors and 24 items, including "emotional support", "informational support", "environmental adjustment support", and "evaluative support", and was developed to visualize the family support practices of ICU nurses and be used as a reflective evaluation by the nurses themselves. The model fit indices for the scale were comparative fit index (CFI) = 0.96 and root mean square error of approximation (RMSEA) = 0.07, and McDonald's $\omega$ coefficient for internal consistency was 0.92. Therefore, its reliability and validity have been confirmed in Japan. With the permission of the creator, this scale was used in a study to evaluate its criterion-related validity. The respondents' answers to each question were used on a six-point scale ranging from "strongly agree" to "completely disagree", to investigate their experiences and thoughts.

### 2.4. Data Analysis

For statistical analysis, an item analysis was performed. Items 6–10 were reverse coded before calculating the means of the total and subscales scores. Exploratory factor analysis (EFA), confirmatory factor analysis (CFA), and criterion-related validity were performed to

assess the scale's validity. To assess the reliability, internal consistency and reproducibility were examined. For all analyses, a *p*-value of <0.05 was considered to be significant. All statistical analyses were performed using SPSS 28.0 software (IBM Japan, Tokyo, Japan).

*2.5. Item Analysis*

In the item analysis, ceiling and floor effects were identified from the standard deviations and means to see the bias in the distribution of the data. The good–poor (G-P) analysis and item–total (I-T) correlations were also detected to examine the validity of each item against the scale.

*2.6. Validity Analysis*

An EFA was conducted to examine the factor structure of QFIFE. The Kaiser–Meyer–Olkin (KMO) test and Bartlett's specificity test were performed to assess sampling adequacy and fit of the data to the factor analysis [16]. Scree plots were utilized to estimate the ideal number of potential factors. Thereafter, we assessed whether the variance explained by each factor had an eigenvalue $\geq 1$ and a KMO > 0.5 [17]. Then, an EFA with maximum likelihood (ML) and Promax rotation was performed. [18]. Items with factor loadings <0.35 were excluded. In general, the total variance explained was >50% [19]. The CFA was used to evaluate the goodness-of-fit of the factor structure of the QFIFE-J. Hoyle [20] and Kline [21] recommend using at least four adjustment indices. Thus, in this study, the chi-square test, CFI, Tucker and Lewis's incremental index (TLI), and RMSEA were used to assess data fitness. The goodness-of-fit of the model was good with CFI > 0.90, TLI $\geq$ 0.90, and RMSEA < 0.08. Moreover, the ratio of chi-squares concerning degrees of freedom $(\chi^2/\text{df} < 2.00)$ was evaluated. For criterion validity, Pearson's correlation coefficient was used to assess the correlation between the proposed scale scores and the Family Assistance Practice Scale for ICU Nurses' scores.

*2.7. Reliability Analysis*

To check the reliability of the scale, Cronbach's $\alpha$ coefficients were calculated for the entire scale and each subscale. To check for reproducibility, test–retests were conducted at two-week intervals, and intraclass correlation coefficients for the QFIFE-J response scores were calculated [13].

*2.8. Ethical Considerations*

It was explained in writing that research cooperation was based on free will, that no disadvantages would be incurred by the participants in the event of refusal, and that protection of personal information and privacy would be observed in the publication of the results. The questionnaire was answered only by those who agreed to participate in the study. Data were collected anonymously using serial IDs, and data confidentiality was maintained. Approval for the above was obtained from the Ethics Committee of the author's university (No. 2021F19).

**3. Results**

*3.1. Participants' Demographic Characteristics*

Participants of this study were 250 nurses in the critical care area; 67.2% were older than 30-years-old, 83.2% were women, 54% had a technical college degree, and 44% had a college degree (Table 1). More than half had more than six years of clinical nursing experience; however, more than half had less than five years of experience in the critical care area. Of the nurses, 15.6% held administrative positions in their departments and only a few were certified nurse specialist (CNS) or had certified nurse licenses. More than two-thirds of participants belonged to hospitals with more than 200 beds.

**Table 1.** Participants' characteristics (N = 250).

| Characteristic | Category | n (%) |
|---|---|---|
| Age (in years) | <25 | 20 (8.0) |
| | 25–29 | 62 (24.8) |
| | 30–49 | 152 (60.8) |
| | >49 | 16 (6.4) |
| Sex | Male | 42 (16.8) |
| | Female | 208 (83.2) |
| Educational background | Vocational school | 135 (54.0) |
| | University | 110 (44.0) |
| | Graduate school | 5 (2.0) |
| Clinical experience (in years) | <1 | 9 (3.6) |
| | 1–5 | 67 (26.8) |
| | 6–15 | 120 (48.0) |
| | >15 | 54 (21.6) |
| ICU experience (in years) | <1 | 44 (17.6) |
| | 1–5 | 117 (46.8) |
| | 6–15 | 82 (32.8) |
| | >15 | 7 (2.8) |
| Certification | Certified nurse specialist | 5 (2.0) |
| | Certified nurse | 6 (2.4) |
| | Registered nurse | 239 (95.6) |
| Position | Nurse manager | 8 (3.2) |
| | Assistant nurse manager | 31 (12.4) |
| | General nurse | 211 (84.4) |
| Number of hospital beds | <200 | 23 (9.2) |
| | 200–300 | 34 (13.6) |
| | 301–500 | 80 (32.0) |
| | 501–800 | 57 (22.8) |
| | >800 | 56 (22.4) |
| Type of unit | Adult ICU | 202 (80.8) |
| | Neonatal ICU | 48 (19.2) |

Abbreviation: ICU, intensive care unit.

### 3.2. Validity and Reliability of the QFIFE-J

Answer Distribution and Item Analysis

Data from 250 critical care nurses were used for item analysis (Table 2). Most items had the highest frequency of "4-point" responses, while Item 9 had the highest frequency of "1-point" responses. No ceiling or floor effects were found as the maximum value of the sum of the means and standard deviations for each item was 5.56 and the minimum value was 1.06. In the G-P analysis, a significant difference between the means of the high- and low-scoring groups was found for each item. The I-T correlations were significantly correlated with the overall scores for all items. Thus, no items were removed as a result of the item analysis.

### 3.3. Validity Testing

To determine the construct validity, item analysis was performed on the 15 items of the proposed scale, and no items were deleted. Therefore, EFA was performed using the ML method and Promax rotation on all 15 items. The KMO index for sampling adequacy was 0.799. Based on eigenvalues ≥ 1.0 and scree plots, four factors were identified. Items with loading values ≥ 0.35 were retained. Finally, a 15-item scale with a four-factor structure was used. To interpret and name each factor, the theoretical characteristics derived from the four-factor structure employed in the EFA were compared with the constructs identified in a previous study. Factor 1 comprised five items (Items 1–5) and was named "ICU

environment". Factor 2 comprised five items (Items 11–15) and was named "nurses' attitudes". Factor 3 had three items (Items 8–10) and was named "nurses' workflow". Factor 4 comprised two items (6 and 7) and was named "patient acuity".

**Table 2.** The QFIFE-J item analysis (N = 250).

| Item | | Mean (SD) | Good–Poor Analysis | | | | Item–Total Correlation Analysis |
|---|---|---|---|---|---|---|---|
| | | | High Mean (SD) | | Low Mean (SD) | | |
| 1 | My unit is physically set up in a way that makes it possible to involve family caregivers in patient care. | 3.32 (1.33) | 3.98 | (1.18) | 2.70 | (1.16) | 0.58 ** |
| 2 | My unit is adequately staffed to allow me time to involve family caregivers in patient care. | 2.94 (1.32) | 3.60 | (1.26) | 2.33 | (1.07) | 0.55 ** |
| 3 | My unit has established written policies for involving family caregivers in patient care. | 2.88 (1.26) | 3.51 | (1.20) | 2.28 | (0.99) | 0.54 ** |
| 4 | My unit supports family caregivers' presence during procedures (e.g., resuscitation, line placement). | 2.41 (1.35) | 2.79 | (1.48) | 2.05 | (1.10) | 0.32 ** |
| 5 | There is a designated space and resources for families who wish to remain with their loved ones in the ICU. | 3.23 (1.25) | 3.78 | (1.11) | 2.71 | (1.14) | 0.52 ** |
| 6 | Family caregivers of patients who are hemodynamically unstable should be excluded from participating in patient care. | 3.86 (1.23) | 4.14 | (1.22) | 3.60 | (1.18) | 0.31 ** |
| 7 | Patients on life-sustaining treatments should not have family caregivers involved in patient care. | 4.36 (1.20) | 4.73 | (1.14) | 4.01 | (1.14) | 0.39 ** |
| 8 | Allowing family caregivers to assist in patient care interrupts my work. | 3.60 (1.13) | 3.93 | (1.10) | 3.30 | (1.08) | 0.38 ** |
| 9 | My clinical performance will be affected by the presence of family caregivers in the room while I am providing patient care. | 3.60 (1.12) | 3.90 | (1.15) | 3.31 | (1.01) | 0.35 ** |
| 10 | I am too busy to incorporate family caregivers in patient care. | 3.80 (1.19) | 4.17 | (1.17) | 3.45 | (1.10) | 0.42 ** |
| 11 | Allowing family caregivers to assist in patient care could help me more accurately assess distressing symptoms in my patients. | 3.76 (0.99) | 4.19 | (0.88) | 3.36 | (0.93) | 0.54 ** |
| 12 | Allowing family caregivers to assist in daily patient care could improve the caregivers' levels of stress, anxiety, and fear. | 4.20 (0.96) | 4.59 | (0.79) | 3.84 | (0.97) | 0.47 ** |
| 13 | I think that family caregivers who engage in patient care are better able to make care decisions for their loved ones. | 4.06 (0.92) | 4.50 | (0.73) | 3.64 | (0.89) | 0.57 ** |
| 14 | I think involving family caregivers in patient care improves patient safety. | 3.51 (1.03) | 3.92 | (0.98) | 3.12 | (0.93) | 0.51 ** |
| 15 | I think involving family caregivers in patient care improves overall quality of care. | 4.17 (1.05) | 4.76 | (0.79) | 3.62 | (0.97) | 0.65 ** |

Abbreviation: ICU, intensive care unit. ** $p < 0.01$.

Factors 1–4 explained 26.533%, 18.308%, 11.689%, and 6.357% of the total variance, respectively, with a cumulative contribution rate of 62.886%. The CFA for this 15-item model

yielded the following indices: $\chi^2(84) = 153.535$, $\chi^2/\text{df} = 1.828$, CFI = 0.942, TLI = 0.917, and RMSEA = 0.057 (Table 3). The results of the criterion-related validity assessment are presented in Table 4. In this assessment, the extent that the total score on the QFIFE-J and each factor correlated with the total score on the Family Assistance Practice Scale for ICU Nurses was examined.

**Table 3.** Results of the exploratory factor analysis and Cronbach's α coefficients (N = 250).

| | Item | Factor Loading | | | | Mean (SD) |
|---|---|---|---|---|---|---|
| | | **1** | **2** | **3** | **4** | |
| | Factor 1: ICU environment (α = 0.813) | | | | | 2.96 (0.98) |
| 3 | My unit has established written policies for involving family caregivers in patient care. | **0.852** | −0.052 | −0.104 | 0.045 | |
| 2 | My unit is adequately staffed to allow me time to involve family caregivers in patient care. | **0.838** | −0.071 | 0.046 | −0.002 | |
| 1 | My unit is physically set up in a way that makes it possible to involve family caregivers in patient care. | **0.724** | 0.059 | −0.065 | 0.076 | |
| 5 | There is a designated space and resources for families who wish to remain with their loved ones in the ICU. | **0.562** | 0.132 | 0.109 | −0.189 | |
| 4 | My unit supports family caregivers' presence during procedures (e.g., resuscitation, line placement). | **0.387** | 0.045 | 0.060 | −0.146 | |
| | Factor 2: Nurses' attitudes (α = 0.794) | | | | | 3.94 (0.73) |
| 12 | Allowing family caregivers to assist in daily patient care could improve the caregivers' levels of stress, anxiety, and fear. | −0.152 | **0.855** | −0.102 | −0.018 | |
| 13 | I think that family caregivers, involved in patient care, are better able to make care decisions for their loved ones. | 0.035 | **0.781** | −0.032 | −0.005 | |
| 15 | I think involving family caregivers in patient care improves overall quality of care. | 0.164 | **0.585** | −0.015 | 0.155 | |
| 11 | Allowing family caregivers to assist in patient care could help me more accurately assess distressing symptoms in my patients. | 0.069 | **0.566** | 0.099 | −0.106 | |
| 14 | I think involving family caregivers in patient care improves patient safety. | 0.065 | **0.400** | 0.135 | 0.101 | |
| | Factor 3: Nurses' workflow (α = 0.687) | | | | | 3.67 (0.90) |
| 8 | Allowing family caregivers to assist in patient care interrupts my work. | −0.026 | 0.010 | **0.882** | −0.048 | |
| 9 | My clinical performance will be affected by the presence of family caregivers in the room while I am providing patient care. | −0.052 | 0.021 | **0.598** | 0.057 | |
| 10 | I am too busy to incorporate family caregivers in patient care. | 0.205 | −0.066 | **0.374** | 0.202 | |
| | Factor 4: Patient acuity (α = 0.780) | | | | | 4.11 (1.10) |
| 7 | Patients on life-sustaining treatments should not have family caregivers involved in patient care. | −0.034 | 0.018 | 0.010 | **0.867** | |
| 6 | Family caregivers of patients who are hemodynamically unstable should be excluded from participating in patient care. | −0.142 | 0.027 | 0.134 | **0.638** | |
| | Factor loading (%) | 26.533 | 18.308 | 11.689 | 6.357 | |
| | Cumulative loading (%) | | 44.840 | 56.529 | 62.886 | |
| | Cronbach's α (full scale) = 0.779 | | | | | |
| Inter-factor correlations | Factor 1 | 1.000 | 0.387 | 0.093 | −0.051 | |
| | Factor 2 | | 1.000 | 0.135 | 0.331 | |
| | Factor 3 | | | 1.000 | 0.482 | |
| | Factor 4 | | | | 1.000 | |

Note. Factor loadings for each item in the factor are shown in bold.

**Table 4.** Relationship between the QFIFE-J and Family Assistance Practice Scale for ICU Nurses (N = 250).

| | | QFIFE-J | | | |
| | Total Score | Factor 1: ICU Environment | Factor 2: Nurses' Attitude | Factor 3: Nurses' Workflow | Factor 4: Patient Acuity |
| --- | --- | --- | --- | --- | --- |
| The Family Assistance Practice Scale for ICU Nurses | | | | | |
| Total score | −0.047 | 0.292 ** | 0.598 ** | −0.848 ** | −0.730 ** |
| Emotional support | −0.157 * | 0.131 * | 0.594 ** | −0.628 ** | −0.648 ** |
| Information provision support | −0.198 ** | 0.262 ** | 0.540 ** | −0.911 ** | −0.814 ** |
| Environmental coordination support | 0.043 | 0.363 ** | 0.511 ** | −0774 ** | −0.563 ** |
| Family support behavior | 0.045 | 0.275 ** | 0.542 ** | −0.665 ** | −0.558 ** |

Note. Pearson's rank correlation coefficient for the total score between QFIFE-J and Family Assistance Practice Scale for ICU Nurses. ** $p < 0.01$, * $p < 0.05$.

Pearson's rank correlation coefficient for the total scores was −0.047; the values for factors 1–4 ranged from −0.848 to 0.598 ($p < 0.01$).

*3.4. Reliability Testing*

The internal consistency analysis of the final four-factor, 15-item scale showed high reliability for all factors, with the Cronbach's $\alpha$ coefficient for the overall scale being 0.779.

The coefficients for the subscales "ICU environment", "nurses' attitudes", "nurses' workflows", and "patient acuity" were 0.813, 0.794, 0.687, and 0.780, respectively. Of the 250 respondents who completed the first survey, 160 (64.0%) participated in the second survey. The intraclass coefficient for test–retest reliability was 0.804 ($p < 0.01$) for the overall scale. The coefficients for the subscales "ICU environment", "nurses' attitudes", "nurses' workflow", and "patient acuity" were 0.762 ($p < 0.01$), 0.567 ($p < 0.01$), 0.587 ($p < 0.01$) and 0.560 ($p < 0.01$), respectively.

**4. Discussion**

In this study, the QFIFE-J was developed using a comprehensive approach. The initial scale items were developed incorporating ICU nurses' values and perspectives on cultural norms, which serve as the bias for tool development in psychometrics [22]. Moreover, the conceptual factor structure (construct validity) was assessed using an EFA. No floor or ceiling effects were found. The results indicated that family support in Japanese ICUs is best explained by a four-factor, 15-item model. Cronbach's alpha was 0.779 for the entire scale, indicating an acceptable internal consistency [23]. For test–retest reliability, the total score of the QFIFE-J showed high reliability. However, some factors did not show high reliability values on the subscales. This is expected because Factor 1, "ICU environment", varied a little; while Factor 2, "Nurses' attitudes", Factor 3, "Nurses' Workflow", and Factor 4, "patient acuity", were those that varied with changes in patient conditions, especially in ICUs. In ICUs, these factors fluctuate with changes in the patient's situation, and it was expected that the content would have changed over the two weeks measured [24]. Regarding construct validity, the EFA results revealed a four-factor construct, as defined in a previous study. Thus, this scale had good reliability and validity and can be considered a useful tool for measuring the barriers and facilitators of family support in the ICU.

Furthermore, the item analysis suggested that the mean scores for items related to nurses' attitudes were generally high, while those for the departmental environment were generally low, thus indicating that, although many nurses have a positive attitude toward involving family members in patient care, they lack organizational support to implement it, and the departmental environment is a challenge for promoting FCC in ICUs in Japan. Thus, it can be said that engaging families in the care of critically ill patients in the ICU requires a more focused team effort based on a shared culture and defined framework of FCC [25], which will require organizational support in the form of hospital and ICU policies

and nurse education designed to promote FCC [4]. However, organizational support to facilitate such FCC in ICUs is lacking and has now become an international hindrance to FCC [26]. The current results show lower ratings for the departmental environment compared to the results in the U.S. [7], thereby highlighting the lack of organizational support as an issue for family care in Japanese ICUs.

The QFIFE-J can measure the perceived barriers and facilitators that affect critical care through nurses' attitudes toward family engagement, during care of critically ill patients. It contains only 15 items that are easy to use in practice. The QFIFE-J can also be used to assess four areas: ICU environment, nurses' attitudes, nurses' workflow and patient acuity, and attitude toward family caregiver engagement in care.

Involving the family in ICU care requires consideration of a variety of influencing factors; therefore, using a scale that takes a broad view may accurately capture the disincentive. In fact, the ability to clarify the facilitating and inhibiting factors in nurses' collaboration with families using this scale may help to accurately describe the current situation in ICUs and effectively collaborate with families. We believe that low scores for each factor of this scale will allow us to consider various measures to improve the factors that hinder collaboration with families.

Regarding the "ICU environment" factor, we should consider relaxing the restrictions on visiting hours and improving the environment for family members to enter the ICU. Regarding the "Nurses' attitudes" factor, we should consider the need for in-hospital family nursing education, i.e., education that enables positive recognition of collaboration with families or education on structured specific communication methods [25]. Regarding the "Nurses' workflow" factor, we should consider staffing, i.e., increasing the number of nurses or making consultation with professionals such as CNS seamless. Regarding the "Patient acuity" factor, we should always try to provide the best possible care to stabilize the patient's condition and carefully give feedback regarding the patient's progress to their family [27]. The recent COVID-19 pandemic has restricted family visits to patients during hospitalization, making it difficult for FCC to take root. It is thus necessary to promote qualitative improvement of nursing care in ICUs in Japan. By utilizing this scale, we believe that it is possible to evaluate what situations and patient attributes make it possible to practice FCC in ICUs in Japan. We also believe that this scale can be used to evaluate nursing education, and it also has implications for family involvement.

The ICU visiting policies vary widely depending on regional factors, cultural differences, laws, and hospital policies [28,29]. As the four subscales of the QFIFE may differ among institutions, systems, and countries, we believe that various comparative studies can provide suggestions for effective collaboration with families.

*Study Limitations*

As this study was conducted online and participants were limited to Rakuten Insight registrants, there was a possibility of selection bias. As the items comprising this scale are Japanese translations of the QFIFE, some items were not necessarily adapted depending on the ICU environment and standards in Japan. Therefore, to assess the facilitators of family care in ICUs in Japan, it is necessary to examine the details of ICU standards and family care practices in Japan, and the findings of nurses and physicians in particular.

## 5. Conclusions

The QFIFE-J is a reliable and valid instrument to quantitatively assess the facilitators and inhibitors of family entry into ICU patient care. Thus, using this scale to identify the facilitating and inhibiting factors in nurses' collaboration with families, it may be possible to accurately understand the current situation in ICUs and effectively collaborate with families.

**Author Contributions:** Conceptualization, M.T. and J.H.; methodology, M.T.; software, Y.I.; validation, M.T. and S.K.; formal analysis, M.T.; investigation, Y.I.; resources, Y.I.; data curation, Y.I.; writing—original draft preparation, M.T. and S.K.; writing—review and editing, M.T., S.K., Y.I., K.S. and J.H.; visualization, S.K.; supervision, J.H.; project administration, M.T.; funding acquisition, M.T. All authors have read and agreed to the published version of the manuscript.

**Funding:** The Foundation of Kinoshita Memorial Enterprise funded this research.

**Institutional Review Board Statement:** This study was approved by the ethics committee of the university to which the authors are affiliated (no. 2021F19).

**Informed Consent Statement:** Informed consent was obtained from all participants involved in the study.

**Data Availability Statement:** All data generated or analyzed during this study are included in this published article.

**Acknowledgments:** The authors are grateful to all study participants.

**Conflicts of Interest:** The authors declare no conflict of interest.

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
