# Peer review of "Development and Validity of the Japanese Version of the Questionnaire on Factors That Influence Family Engagement in Acute Care Settings"

_nursrep, doi:10.3390/nursrep13020053_

Round 1

Reviewer 1 Report

ID: nursrep-2202391

Title: Development and Validity of the Japanese Version of the Questionnaire on Factors that Influence Family Engagement (QFIFE-J)

Thank you for providing a chance to review this manuscript.

Comment: Rejection or Major Revision.

Detailed information:

Title: It is recommended to highlight the research object of this study in the title.

Abstract

Line 17-22, Page 1: This section mainly describes the background of the research, please simplify it.

Line 27-31, Page 1: The results are an important part of the abstract. Please expand and describe the results in detail, including but not limited to statistical results of reliability and validity.

Introduction

1) Please explain the meaning and background of Post-intensive Care Syndrome (PICS) and the Post intensive Care Syndrome-Family PICS-F.

2) Please introduce the background and development of QFIFE-J. In addition, the research basis of QFIFE-J also needs to be elaborated.

3) What was the basis for this study? And what were the assumptions of this study?

Overall: Introduction should provide background/research gap/research objectives, but the author took a lot of part to explain background. This part is too brief, the authors omit some information, please add in detail. Please rewrite the introduction!

Materials and Methods

Design

How can bias in online surveys be controlled?

Materials

1) All questionnaire data need to be supported by relevant literatures, please add.

2) Has the cross-cultural adaptation of QFIFE-J been assessed by the expert group? And did you conduct a pre-experiment?

Measures

Line 96-97, Page 3: “Approximately 6,200 nurses working in ICUs are registered withRakuten Insight, Inc.” With so many registrants, why did only 250 nurses end up recruiting?

Line 100-102, Page 3: “To validate a survey, a sample size that is at least 7 times 100 larger than the number of items it contains and an absolute number of at least 100 are required”. Despite the authors' explanation, more authoritative evidences are needed to show the sample size of this study is adequate. In addition, for cross-cultural validation of QFIFE-J, I think the sample size of this study is far from enough.

Validity Analysis

Line 161, Page 4: For continuous variables, Pearson correlation analysis should be used, not Spearman. Please re-examine.

Results

Line 180-181, Page 4: “The participants were 250 nurses in the critical care area; 67.2% were over the age of 30 83.2% were female” The punctuation mark is missing after 30.

Line 220-222, Page 7: “This assessment examined the extent to which the total score on the QFIFE-J and each factor correlated with the total score on The Family Assistance Practice Scale for ICU Nurses.” There is no correlation between the total scores of the two questionnaires. Please explain.

Line 236-239, Page 9: The intraclass coefficient for test–retest reliability of the overall scale is strong, but the subscales are not good.

Discussion

Line 247-249, Page 9: “To the best of our knowledge, this scale has previously been used to assess family support in ICUs in Japan. Cronbach's alpha value was 0.779 for the entire scale, indicating adequate reliability” This study is not the first Japanese version of QFIFE-J?

Overall: The authors spend most of space repeating the results of this study without elaborating. Please read similar studies and compare the conclusions of other investigators. Also, since this is not the first Japanese version, please explain where this study is innovative.

With all due respect, I did not even finish my reading. I am feeling that there's a lot of information missing here. First, the introduction and discussion sections are badly written. Authors need to read high-quality articles and reorganize the main points and ideas of this paper. Furthermore, there are many basic errors in this article, such as grammar, punctuation and so on. Last but not least, finding a native English speaker to improve the writing can considerably improve the quality.

Thank you and my best,

Your reviewer

Reviewer 2 Report

Nice Job.

It would have been interesting to detail the process of transcultural adaptation of the instrument, which is mentioned in one of the sections but is not detailed. lines 82 to 91 of the manuscript.

This is a work before the validation process of the questionnaire since it is delivered and translated into Japanese to the participants.

In this step, relevant information may have been lost from the original English version to the Japanese translated version.

The cross-cultural adaptation process ensures the minimum loss of content. For this reason, an explanation of the process would have been desirable.

Reviewer 3 Report

This study can contribute to depth knowledge about this topic. This manuscript is clear, relevant for the nursing field and other knowledge fields and are presented in a well-structured manner.  However, there is a need for a more clarification of the implications to practice (discussion) and improve the references.

Line 113 vs line 182 table 1. Refers to sex and gender. Please clarify the concept.

Line 270 – this is an academic work. What is the value to practice? It is important to apply in all ICU contexts? When a new nurse integrates the team? What is the recommendation to use in practice? When should apply this questionnaire?

Line 325 – 16 of the 27 references have more than 5 years. Can you improve this item?

Round 2

Reviewer 1 Report

ID: nursrep-2202391

Title: Development and Validity of the Japanese Version of the Questionnaire on Factors that Influence Family Engagement in Acute Care Settings

The authors have made some modifications to the manuscript according to the previous comments, but there are still some problems that have not been effectively modified. The main problem of this paper is still in the methods and results. Please make further modifications. In addition, the revision mode makes the modification of the article look messy. It is recommended that the author highlight the color of the modified part.

Comment: Major Revision.

Detailed information:

Introduction

The background and research basis of QFIFE-J should be supplemented in the paper.

Materials and Methods

QFIFE-J

The authors add the process of cross-cultural translation. There are still some questions, and authors are advised to refer to the COSMIN guide. Has a panel of experts been invited to review the QFIFE-J? In addition, was the questionnaire pre-tested at the patient level, including focus groups and one-on-one interviews? These are all important elements for cross-cultural adaptation of a questionnaire.

Measures

Although there is no unified consensus on cross-cultural adaptation at present, the sample size used in the verification of cross-cultural adaptation is huge in many studies.

Results

Table 2: Although the Item–Total correlation analysis was statistically significant, the r-values were only between 0.3 and 0.6, which could be considered as not strong correlation.

Discussion

In the discussion, I would prefer to see the comparison between the conclusions of this study and other similar studies, including supporting and contradictory evidence.

Thank you and my best,

Your reviewer
